# Attitudes and Behavior toward COVID-19 Vaccination in Japanese University Students: A Cross-Sectional Study

**DOI:** 10.3390/vaccines10060863

**Published:** 2022-05-27

**Authors:** Yuri Okamoto, Toru Hiyama, Yoshie Miyake, Atsuo Yoshino, Shunsuke Miyauchi, Junko Tanaka

**Affiliations:** 1Health Service Center, Hiroshima University, 1-7-1 Kagamiyama, Higashihiroshima-city, Hiroshima 739-8514, Japan; tohiyama@hiroshima-u.ac.jp (T.H.); miyakechare@hiroshima-u.ac.jp (Y.M.); yoshino@hiroshima-u.ac.jp (A.Y.); smiyauchi@hiroshima-u.ac.jp (S.M.); 2Department of Epidemiology, Infectious Disease Control and Prevention, Graduate School of Biomedical and Health Science, Hiroshima University, 1-2-3 Kasumi, Minami-ku, Hiroshima 734-8553, Japan; jun-tanaka@hiroshima-u.ac.jp

**Keywords:** vaccination, COVID-19, attitude, behavior, college students

## Abstract

Background: It is said that safe and effective vaccination is an important tool to end the COVID-19 pandemic. However, recent studies have reported hesitation, especially in young adults. Promoting the vaccination of university students, who represent the young adults, will lead to infection prevention measures. The purpose of this study was to clarify to compare the vaccination rates, attitudes toward vaccines, and post-vaccination behavior of students and faculty members in order to understand the actual situation of young population. Methods: We conducted large-scale vaccination of Hiroshima University from 21 June to 18 September 2021. This cross-sectional survey was conducted via e-mail from 27 September to 3 October 2021. Results: The number of second inoculations was 10,833 /14,154 students (76.5%), and 2240/2583 staff members (86.7%). Regarding the impressions after vaccination, the most common answer was “I was able to prevent worsening of the disease even if I was infected”. Many students answered that their range of activities had expanded after vaccination. However, many students (*n* = 1799, 87.8%) answered as having “no change after vaccination” regarding infection prevention. Conclusion: The high vaccination rate in this survey was thought to be due to the increased sense of security and confidence in the vaccine. The fact that young adults who perform a wide range of activities are careful about infection prevention may be one of the factors that prevents the explosive spread of infection in Japan.

## 1. Introduction

The WHO Emergency Committee raised international concerns on 23 January 2020, regarding the outbreak of the new coronavirus disease 2019 in the People’s Republic of China [1], and characterized the outbreak of COVID-19 as a pandemic [2]. As of February 2022, more than 402,000,000 confirmed cases of COVID-19 worldwide have been reported to the WHO, including more than 5,700,000 deaths [3]. In Japan, as of February 2022, nearly 34,000,000 cases of COVID-19 have been reported, including nearly 20,000 deaths [4]. The Japanese population were instructed to take thorough infection prevention measures, such as wearing masks, washing hands, avoiding crowds, and promoting ventilation.

It is said that safe and effective vaccination is an important tool to end the COVID-19 pandemic. However, there have been reports of concerns about the public’s awareness of vaccines and the acceptability and intent of vaccination [5,6]. Recent studies have reported hesitation, especially in young adults [5,7]. Several factors contribute to vaccine hesitation, including the fear of potential side effects, previous experiences with health care providers, peer social acceptance, beliefs in conspiracy theories, and concerns [5,7,8]. University students are an important group of young adults that is eligible for COVID-19 vaccination. However, the hesitation of university students to receive vaccines is also regarded as a problem [9], and it seems that promoting the vaccination of university students will lead to infection prevention measures.

Especially regarding acceptancy of booster dose, Yadate et al. [10] report that among 2138 participants, nearly 62% intended to take booster doses and those remaining were hesitant. Furthermore, they reported that a quarter of unvaccinated participants (28.6%) and 13% of vaccinated participants were unsure whether they would accept a booster dose. Yadate et al. [10] indicated that this suggests that there are many individuals who have no fixed opinion on booster shots yet and may be persuadable. We hypothesized that providing accurate information and supporting the concerns of a hesitant group of college students and faculty members in the mass vaccination program would help promote vaccination.

In Japan, free vaccination started last year, and the rate of two vaccinations is 77.5% [11]. The Japanese government has recommended mass vaccination at universities to promote vaccination of young people. In response, we have started a large-scale vaccination program at Hiroshima University. Our Health Service Center is a clinic staffed by psychiatrists, internists, psychologists, and nurses who provide consultation and medical care to students, faculty staff. We conducted large-scale vaccination for students and faculty members of Hiroshima University from 21 June to 18 September 2021 using the Moderna vaccine. Hiroshima University is in the western part of Japan, and Hiroshima Prefecture, where the university is located, has a medium-sized population in Japan. False information about the vaccine was a reason for confusion before vaccination, and there was concern about hesitations in receiving the vaccine. Therefore, we publicized accurate information about the vaccine on the University and Health Service Center websites from 1 June to 18 September and when we sent vaccine reservation forms to all constituents. In addition, information on adverse reactions to vaccines was also provided at the time of vaccination, and contact information (Health Service Center) was indicated in case of adverse reactions. There were many adverse reactions among the students after their first vaccination, and it was speculated that the fear of the vaccine might have increased. We (medical doctors) were consulted by telephone 24 h a day for the first week after the vaccination. As a result, it was speculated that many persons were relieved after vaccination. In order to promote vaccination, it is important to investigate the views, attitudes, and behaviors of university faculty and students toward vaccination. In particular, exploring the attitudes of college students, who are representative of young people, in comparison with faculty and staff will lead to future infection prevention measures.

In addition to vaccination, other recommended infection prevention measures include preventing contact with infected persons by maintaining a physical distance, thorough hand washing, and proper wearing of masks [12,13,14,15]. In particular, wearing masks has been found to significantly reduce COVID-19, and a synergistic effect is expected when combined with vaccination [12]. Therefore, we think that it is important to take infection prevention measures such as wearing masks even after vaccination, and we informed university students of the need for infection prevention measures after vaccination.

The purpose of this study was to compare the vaccination rates, attitudes toward vaccines, and post-vaccination behavior of students and faculty members in order to understand the actual situation of young population. We hypothesized that college students had lower vaccination rates than faculty and staff, that they were more anxious about vaccines, and that they were more active and had more human contact after vaccination.

## 2. Materials and Methods

### 2.1. Participants

This cross-sectional survey was conducted via e-mail from 27 September 2021 to 3 October 2021. All students and faculty members of Hiroshima University were sent an e-mail, providing access to the survey form, and requesting their cooperation in the survey. The advantages of this study were that the two vaccinations were given in the same environment in a mass vaccination program at a university, the information could be disseminated to the entire population, adverse reactions could be easily handled, and data could be collected quickly. The target population consisted of all vaccinators other than those vaccinated at large with the Modena vaccine.

This survey system was designed to allow respondents to access and respond to the survey instrument through a personal e-mail request, and each respondent could only respond once, thus preventing duplicate responses.

### 2.2. Ethical Considerations

This study was conducted in accordance with the guidelines proposed in the Declaration of Helsinki and was approved by the Epidemiology Ethics Committee of Hiroshima University (approval ID: E2123-2, 27 July 2021). When we sent the survey by personal e-mail and requested their cooperation, we told them that their responses would be handled anonymously and that they would not be disadvantaged if they did not cooperate with the survey. It was assumed that consent was obtained by cooperating in the survey.

### 2.3. Measures

The instrument consisted of 16 items about attributes, whether the participants were vaccinated, why they were not vaccinated, why they received only one vaccination, changes in their attitudes and behaviors after vaccination. Questions about the psychological situation included the presence or absence of fear of the vaccine, changes in their fear after vaccination, their impression of vaccination (multiple-choice answer) and whether their anxiety about new coronavirus infections changed after vaccination. The questions about changes in behavior and cognition after vaccination had four answer choices, “quite changed”, “slightly changed”, “not much changed”, and “not changed at all”, regarding whether their understanding and behaviors had changed after vaccination. In addition, after vaccination, we asked if the participants could act with more peace of mind than before vaccination, if there was a change in their contact opportunities with population, if there was a change in their infection prevention measures, and how they felt about the effectiveness of the vaccine. The questionnaire was constructed by collecting the opinions of internists, psychiatrists, public health experts, university administrators, and faculty members.

### 2.4. Statistical Analysis

When asked if their understanding or behaviors had changed before and after vaccination, if a participant answered, “significantly changed” or “slightly changed”, the participant was determined that they had changed their understanding or behavior. If a participant answered, “not much” or “not at all”, it was determined that their understanding and behavior had not changed. The participants were determined to “believe” the effectiveness of the vaccine if they answered “fully” or “somewhat believe” in the effectiveness of the vaccine. Fear and trust in vaccines, anxiety about infection, and changes in awareness and behaviors after vaccination were compared between students (young adults) and faculty staff and tested using the chi-square test for vaccine. Statistical analysis was performed using JMP Pro for Mac version 16 (SAS Institute Japan, Tokyo, Japan). A two-sided *p* value less than 0.05 was considered statistically significant.

For sample size validity, a power analysis was conducted using G*Power software 3.1.9.6 for Mac OS X (Faul, F., et al., Heinrich Heine Universität, Düsseldorf, Germany) [16,17]. The maximum required sample size was 1979 when analyzed with a small effect size of 0.1, alpha level of 0.2, and power of 0.95 required for the chi-square test. The sample size of this study was meet that requirement.

## 3. Results

The number of first inoculations was 10,948/14,154 for students (77.3%), and 2247/2583 for staff members (87.0%); the number of second inoculations was 10,833/14,154 students (76.5%), and 2240/2583 staff members (86.7%).

A total of 2160 students (response rate 13.9%) and 2289 staff members (response rate 62.5%) responded to the survey. There were 101 nonvaccinated persons (78 students, 23 staff members) and 26 persons who received only one vaccination (22 students, four staff members) (Table 1). The reasons for not receiving a vaccination are shown in Figure 1. The most common answer was “because of the fear of side effects”, with 71 respondents (70.2%). Of the 26 persons who had not received a second vaccination, the most common reason was because they “planned to receive the vaccine in the future (the schedule did not match)”, with 19 respondents (73.1%).

### 3.1. Analysis of Students/Faculty Staff Who Have Been Vaccinated Twice

#### 3.1.1. Fear of Vaccination before and after Vaccination

Figure 2 shows the fear of vaccination before and after vaccination. For students, 842 (41.1%) had a fear of the vaccine before vaccination. Of these, 151 (17.9%) had a feeling of fear even after vaccination, and 471 (55.9%) had no feelings of fear. Similarly, for faculty and staff members, 950 (42.1%) were afraid of vaccination before receiving a vaccination. Of these, 298 (31.4%) were still afraid after receiving a vaccination, and 382 (40.2%) were not afraid. Figure 3 shows a comparison between students and faculty staff regarding their fear of vaccine after vaccination. Significantly more students had no fear after vaccination (χ^2^(2) = 54.3, *p* < 0.0001).

#### 3.1.2. Impressions about Vaccination

Figure 4 shows the respondents’ impressions after vaccination. For both students and faculty members, the most common answer was “I was able to prevent worsening of the disease even if I was infected”. The second most common answer was “I was able to contribute to the prevention of infection”, and the third was “I was relieved by receiving the vaccination.”

### 3.2. Comparison of Students and Faculty and Staff

There was no difference between the students and faculty and staff regarding the trust in vaccination. Many students and faculty and staff members answered that they believed in the vaccine completely/to some extent (Table 2).

Table 3 and Table 4 show a comparison of the students and faculty and staff regarding changes in their attitudes and behaviors after vaccination. The number of students whose understanding and behaviors changed was significantly higher than that of the faculty and staff members (χ^2^(1) = 16.8, *p* < 0.0001), and the number of students who answered that they could act with peace of mind was significantly higher (χ^2^(2) = 129.7, *p* < 0.0001). In addition, significantly more students answered that their range of activities had expanded (χ^2^(3) = 133.4, *p* < 0.0001) and that they had more opportunities to interact with population after vaccination (χ^2^(3) = 59.5, *p* < 0.0001) than faculty staff. Regarding infection prevention, the students who answered “little to being careful” scored significantly higher than the faculty and staff members (χ^2^(3) = 126.3, *p* < 0.0001). However, many students (*n* = 1799, 87.8%) and many faculty and staff members (*n* = 2066, 91.5%) answered that they had “no change after vaccination”.

## 4. Discussion

With the spread of COVID-19 and the emergence of mutant strains, hesitation to receive the vaccination is a concern worldwide. A rapid systematic review of 23 peer-reviewed studies and 103 additional studies on the hesitation to receive the COVID-19 vaccine in the United States and around the world showed that the factors included a perceived risk, concerns about vaccine safety and efficacy, and physicians. It has been shown that recommendations and vaccination history influence vaccine hesitation [17]. Studies have suggested that young adults are more likely to hesitate to receive the vaccine than those in other age groups [18,19]. Japan is said to be one of the least reliable countries in the world for vaccines [20], but a survey of Japanese adults [21] reported that 62.1% of adults were willing to receive the vaccine when it became available. In our university-wide, large-scale vaccination program, 86.7% of staff and 76.5% of university students received two vaccinations. The reason for the high vaccination rate is related to the fact that we, the Health Service Center who conducted the vaccination program, also carried out the following: disseminated information about vaccines, disseminated information about COVID-19, mental health care, such as how to treat anxiety, and consultation when infected, and discussed side effects and reactions after vaccination. It is thought that the dissemination of information and continued wide-ranging support have increased the sense of security and confidence in the vaccine. Khubchandani et al. [22] and Yadate et al. [10] report individuals with high level educations such as university students were not likely hesitant about the COVID-19 vaccine. However, many social factors, such as race, employment status, and income, are said to play a role in vaccine hesitancy, and further research is needed.

Some studies [23,24,25] showed that acceptors had positive consequences of receiving the COVID-19 vaccine more often than non-acceptors. This includes (1) reducing the risk of COVID-19 infection, (2) being able to participate in social and cultural activities, (3) being able to reopen children’s schools, (4) reducing COVID-19-related costs, (5) increased employment and income opportunities, and (6) participation in group prayers. Conversely, nonrecipients who were asked about the adverse effects of receiving the COVID-19 vaccine said they would have (1) life-threatening side effects, (2) the development of unknown/new illnesses, and (3) infertility. These effects were likely to be mentioned as a disadvantage. In this study, 70.2% of those who did not receive the vaccine cited the fear of side effects as the reason. The fact that some persons were afraid even though they were disseminating information about side effects requires the consideration of a method for disseminating information. According to a systematic review [26], regarding vaccine hesitation, health care provider communication and behaviors strongly influence patient acceptability and uptake. It seemed important for us as medical personnel to provide accurate information and actively communicate to treat anxiety.

In a study of university students’ vaccine-accepting behaviors [27], only 50% reported that health care providers recommended COVID-19 vaccination. Young adults who hesitated to receive the vaccine were reported to have low beliefs about the benefits of the vaccine, indicating the need for a health promotion program [28]. Designs for sending messages regarding vaccination need to be considered, and health care providers seem to play a powerful role in this. As an on-campus medical center, we were able to reduce the anxiety of university students after the vaccination by posting information on our website, directly engaging in interviews and first aid at the vaccination site, and providing 24-h post-vaccination consultation services.

Young adults often obtain information from social media. A study [18] found that TV channels (63.2%) and social media (53.6%) were the main sources of information related to COVID-19, followed by official sites (45.6%) and family and friends (42.3%). Many Japanese college students obtain information from the internet rather than from TV. When they search for an internet service, much information related to the search will be provided for the next time. Therefore, if a large amount of incorrect information is provided, there is a concern that the opportunity for correction may be lost. To prevent the risk of being brainwashed by incorrect information, it is necessary to devise opportunities and methods to provide sufficient accurate information. Regarding the result of this study, more students had less fear of vaccination than staff members after vaccination. It seems that our information dissemination, et al., as mentioned above were effective. It is important to widely disseminate such facts to young adults.

Regarding impressions after vaccination, many students and faculty members commented that “I was able to prevent worsening of the disease even if I was infected”, “I was able to contribute to the prevention of infection”, and “I was relieved by receiving the vaccination”. Some studies [29] suggested that over 80% of young adults engage in preventive behaviors that are mainly motivated by social responsibility (78.1%) and wanting to protect others (77.9%) rather than personal perceived risk (57.8%). In this study, many university students stated that they were able to contribute to society through vaccination, and it was found that they had a strong sense of social responsibility. This result suggests the importance of educational programs for young adults.

In this study, many students and faculty and staff members answered that they believed in the vaccine completely/to some extent. This may be the result of accurate information dissemination in advance. By accurately providing positive and negative information, such as the effects and side effects of vaccines, we may have gained the trust of the university students and staff members.

Many students answered that their range of activities had expanded significantly after vaccination and that they had more opportunities to interact with population. However, many students (*n* = 1799, 87.8%) and many faculty and staff members (*n* = 2066, 91.5%) answered that they had “no change after vaccination”. It was found that the students were careful to prevent infection, even though vaccination expanded their contact opportunities and range of activities. Even before the spread of COVID-19, many of the Japanese population had many opportunities to wear masks, and many did not feel reluctant to wear them. Furthermore, the fact that young adults with a wide range of activities are careful about infection prevention may be one of the factors that prevents the explosive spread of infection in Japan. A South Korean study [30] used mathematical modeling to show that the changes in social distancing and public behaviors suppressed the spread of COVID-19, demonstrating the importance of these measures in reducing the incidence of infection. College students, an important group of young adults, are said to be vulnerable to COVID-19 infection due to various factors, such as widespread activities both on and off campus and frequent interactions with population. It is thought that many young adults can contribute to the prevention of the spread of infection by taking thorough infection prevention measures along with vaccination. Siddiqui et al. [31] reported three types of protective behaviors: the physical level measures (personal hygiene); the social level measures (physical distancing as directed by the government and healthcare professionals); and the third level measures (religious coping, which is a cognitive reappraisal of the stressful event, human limitations, and corresponding religious beliefs). The present study reveals that preventive measures are taken at the social level as well as at the level of individual students. In addition, several reports [31,32,33,34] have stated that responsible behavior and perceived risk are associated with infection prevention behavior. In the present study, students stated that they were able to “contribute to society” after vaccination, suggesting that such a sense of social responsibility was associated not only with vaccination but also with subsequent persistence of infection-prevention behavior. Tang et al. [35] and Jose et al. [33] stated that individual prevention activities are important to prevent the spread of infection, and that a primary health care approach is important. Zhong et al. [36] stated that infection prevention behaviors are associated with reduced anxiety. We need to consider effective strategies for further health promotion for university students, including dissemination of information such as these reports.

One of the limitations of this research is that the survey was conducted at one local university. Another limitation is that the survey was given immediately after completing two large-scale vaccination programs. More important suggestions may be obtained if we can consider whether there is a difference depending on the situation of the spread of infection and whether there is a change after the end of the third infection outbreak. Furthermore, since this study was conducted only once, it would be helpful to understand the psychological and behavioral changes after a longer period of time or after the third vaccination. There was also the possibility that a response bias existed in that those who had been vaccinated were more likely to respond. Future surveys should be devised to address this bias.

## 5. Conclusions

The high vaccination rate in this survey was thought to be due to the increased sense of security and confidence in the vaccine through the dissemination of information and continuous wide-ranging support. The fact that young adults who perform a wide range of activities are careful about infection prevention may be one of the factors that prevents the explosive spread of infection in Japan. In the future, it is necessary to further consider effective strategies for improving the health of university students.

## Figures and Tables

**Figure 1 vaccines-10-00863-f001:**
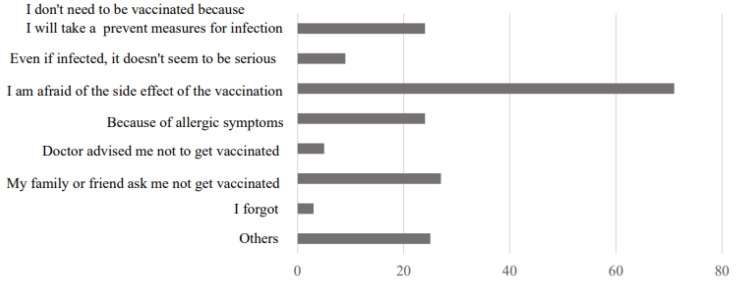
Reasons for not vaccination (multiple answers).

**Figure 2 vaccines-10-00863-f002:**
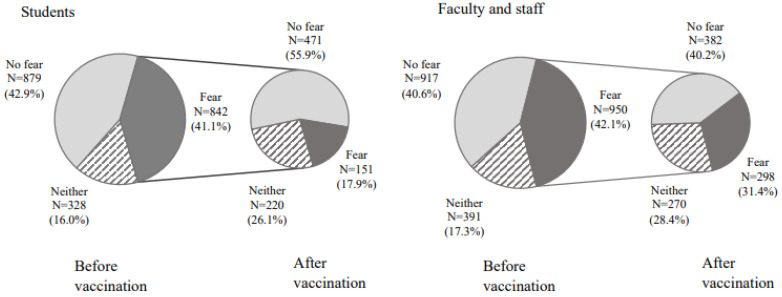
Fear of vaccine before and after vaccination.

**Figure 3 vaccines-10-00863-f003:**
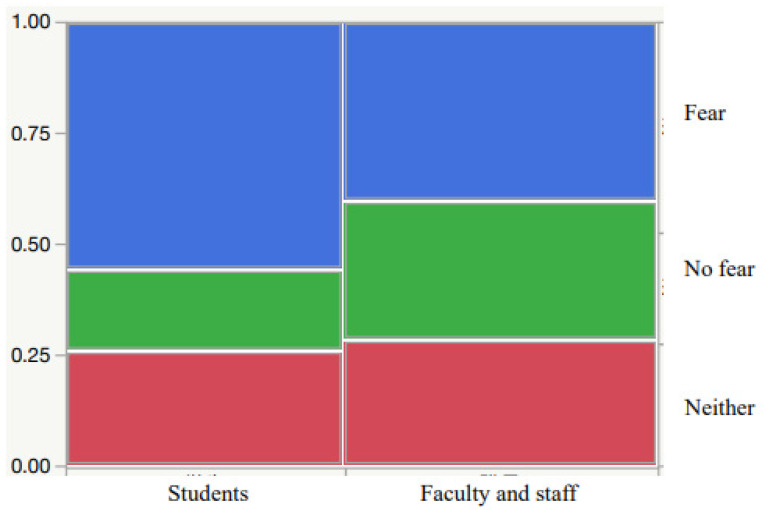
Fear of vaccine after vaccination. Figure shows a comparison between students and faculty staff regarding their fear of vaccine after vaccination. Significantly more students had no fear after vaccination.

**Figure 4 vaccines-10-00863-f004:**
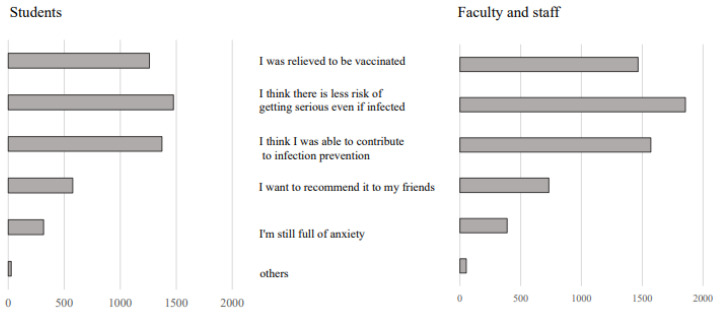
Impressions after vaccination.

**Table 1 vaccines-10-00863-t001:** Study population.

Demography	Gender	Unvaccinated	One Vaccination	Two Vaccinations	Total
Students	Male	42	11	1000	1053
	Female	35	11	1049	1095
	Neither	1	0	11	12
Total		78	22	2060	2160
Faculty and staff	Male	5	2	1096	1103
	Female	18	2	1162	1182
	neither	0	0	4	4
Total		23	4	2262	2289

**Table 2 vaccines-10-00863-t002:** Comparison of students and faculty and staff (fear and trust in vaccination).

		Students N (%)	Faculty and Staff N (%)
Gender	Male	1000 (48.8)	1096 (48.5)
	Female	1049 (51.2)	1162 (51.5)
Fear of vaccination (before vaccination)	Yes	842 (41.1)	950 (42.1)
	No	879 (42.9)	917 (40.6)
	Neither	328 (16.0)	391 (17.3)
Trust in vaccination (before vaccination)	I totally believe	164 (8.0)	205 (9.1)
	I believe to some extent	1664 (81.2)	1819 (80.6)
	I don‘t believe much	94 (4.6)	82 (3.6)
	Neither	127 (6.2)	152 (6.7)

**Table 3 vaccines-10-00863-t003:** Comparison of students and faculty and staff.

		Students N (%)	Faculty and Staff N (%)
Changes in feelings and behavior after vaccination	Yes	567 (27.7) **	503 (22.3)
	No	1482 (72.3)	1755 (77.7) **
Act with peace of mind after vaccination	Yes	1141(55.7) **	866 (38.4)
	No	282 (13.8)	426 (18.8) **
	Neither	626 (30.5)	966 (42.8) **
Anxiety about infection after vaccination	Increased	16 (0.8)	20 (0.9)
	Reduced	822 (40.1)	981 (43.4)
	No change	1154 (56.3)	1217 (54.0)
	I have no anxiety from before	57 (2.8)	40 (1.8)

** *p* < 0.01.

**Table 4 vaccines-10-00863-t004:** Comparison of students and faculty and staff.

		Students N (%)	Faculty and Staff N (%)
Changes in range of action after vaccination	Extended	271 (13.2) **	87 (3.9)
	Narrowed	6 (0.3)	21 (0.9)
	No change	1770 (86.4)	2140 (94.8)
	Neither	2 (0.1)	10 (0.4)
Contact with people after vaccination	Increased	214 (10.4) **	53 (2.4)
	Reduced	16 (0.8)	33 (1.5) **
	No change	1818 (88.7)	2167 (96.0)
	Neither	1 (0.1)	5 (0.2)
About infection prevention after vaccination	Be more careful	97 (4.7)	115 (5.1)
	Little to be careful	117 (5.7) **	32 (1.4)
	No change	1799 (87.8)	2066 (91.5) **
	Neither	36 (1.8)	45 (2.0)

** *p* < 0.01.

## Data Availability

Not applicable.

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
