# Peer review of "Attitudes and Behavior toward COVID-19 Vaccination in Japanese University Students: A Cross-Sectional Study"

_vaccines, 2022, doi:10.3390/vaccines10060863_

Round 1

Reviewer 1 Report

Current and cohesive manuscript

Method with scientific and statistical rigor

Results relevant to the findings

Appropriate and congruent discussion

Conclusion responds to the object of investigation

Author Response

Dear Reviewer,

Thank you for your warm comments. We have taken them to heart. It will be an encouragement to us in the future.

We have revised the manuscript in response to the comments of the other reviewers. The revised parts are in red.

Reviewer 2 Report

The study presented survey results among students and staff at Hiroshima University before and after vaccination for COVID-19. The vaccination program itself was remarkably successful. I suggest the following revisions:

1) There is a consistent issue beginning in the title and throughout the paper where the language would be confusing to a casual reader. The title will sound to many native English speakers as if the authors are suggesting that vaccination biologically changed students psychology or cognition. This is of course not what they are actually reporting. Please work closely with the editors or others to update the title and text so that it is clear you are referring to changing attitudes about the vaccine itself before and after vaccination. 

2) It is not clear to me at what time points the survey was given. I believe it went out only once, with students and faculty being asked about changes. It would have been better if the students and faculty were surveyed before and after with a series of similar questions.  That not being the case, how does issues with self-reporting change in beliefs and behaviors at once time point influence your study?

3) The authors report that the vaccination program itself was so successful because the University disseminated good information. How was the timing of the educational information related to the timing of the survey?

4) I suspect participation bias in the survey, with students and faculty who were vaccinated being more likely to respond. Please address participation bias in your methodologies. 

5) The tables need clarification, including in the legends. For example, the legend for Table 3 reads "Comparison of students and faculty and stuff".(Obviously there is a typo where "stuff" should be "staff". What is being compared is not clear in the tables, as it suggests it is comparing change, but the tables themselves don't clearly show a "before and after".  

Author Response

Dear Reviewer,

Thak you for your insightful comments on our manuscript. We feel the comments have helped us significantly improve the manuscript. 

Point 1: There is a consistent issue beginning in the title and throughout the paper where the language would be confusing to a casual reader. The title will sound to many native English speakers as if the authors are suggesting that vaccination biologically changed students psychology or cognition. This is of course not what they are actually reporting. Please work closely with the editors or others to update the title and text so that it is clear you are referring to changing attitudes about the vaccine itself before and after vaccination. 

Response 1: Indeed, as you pointed out, the wording was meant to confuse the reader.

We changed the title to “Attitudes and behaviors toward COVID-19 vaccination in Japanese university students” and clearly stated that the purpose of the study was to grasp the actual situation of university students by comparing their attitudes and behaviors after vaccination with those of faculty members. ( Lines 1, 19-24, 103-108)

Point 2: It is not clear to me at what time points the survey was given. I believe it went out only once, with students and faculty being asked about changes. It would have been better if the students and faculty were surveyed before and after with a series of similar questions.  That not being the case, how does issues with self-reporting change in beliefs and behaviors at once time point influence your study?

Response 2: There was only one survey. I agree with your point that the actual situation could have been better understood if the survey had been conducted on a continuous basis. Therefore, I have included this as a future issue in the Limitations section of this study. (Lines 336-340)

Point 3: The authors report that the vaccination program itself was so successful because the University disseminated good information. How was the timing of the educational information related to the timing of the survey?

Response 3: We have added an explanation of the timing of our information dissemination and how we can help. (Lines 81-86, 88-94, 265-268)

Point 4: I suspect participation bias in the survey, with students and faculty who were vaccinated being more likely to respond. Please address participation bias in your methodologies.

Response 4: In this survey, we (1) surveyed the vaccination rate, (2) surveyed those who had not been vaccinated about their reasons for not vaccinating, and (3) surveyed only those who had been vaccinated twice about their attitudes and behaviors after vaccination. The survey on attitudes and behavior after vaccination was conducted only among those who had been vaccinated twice, so it is an independent survey. However, certainly, there is a bias that those who have been vaccinated are more likely to respond. The points raised were described as issues for this study. (Lines 338-340)

Point 5: The tables need clarification, including in the legends. For example, the legend for Table 3 reads "Comparison of students and faculty and stuff".(Obviously there is a typo where "stuff" should be "staff". What is being compared is not clear in the tables, as it suggests it is comparing change, but the tables themselves don't clearly show a "before and after".

Response 5: Thank you for pointing this out. Since Tables 2 and 3 are comparisons between students and faculty and staff, I titled them "Comparison between Students and Faculty and staff" and specified that the comparison item is post-vaccination attitudes and behaviors. (Table 2,3) We added Figure 3.

Reviewer 3 Report

Reviewer comments for authors

Thank you for the opportunity to review the manuscript titled “Psychological and behavioral changes in Japanese college students after COVID-19 vaccination.” The findings of this study will help in designing targeted interventions to promote vaccine uptake students. Despite this contribution, I believe there are opportunities to further strengthen this manuscript. I will share these in the order that they appear.

  1. Title: Title of the study does not reflect its design. Authors may consider adding information about the study design.

  1. Introduction
  • The introduction section not back up by sufficient literature review and is superficial. In other words, it does not cover the entire breadth of the issue at hand. Relevant references can be added to strengthen the evidence related to the factors associated with vaccine hesitancy. Tesfaye et al., in one of the articles (given below) emphasize on vaccine literacy and vaccine confidence index.

             https://www.mdpi.com/2076-393X/9/12/1424

             https://pubmed.ncbi.nlm.nih.gov/33389421/

  • Authors are encouraged to include some information related to the uptake of other protective behaviors e.g. handwashing and mask wearing by the younger population.
  1. Methods

Section 2.1: First sentence needs to be re-written. If internet survey is used, then by default the cross-sectional study was done electronically. There is no need to emphasize this.

Please re-write line # 67.

Please add the date of ethical approval.

Please move the ethical information into a separate paragraph and label it as “Ethical Considerations.” In addition to the Ethical approval, what others measures authors have taken to ensure the ethical conduct of the study?

How authors prevented multiple responses from the same participant?

Which platform was used to collect the electronic data?

Was the survey tool validated? It is critical to use valid instruments.

Statistical analysis portion does not provide the information on the power analysis and type of statistical tools other than Chi-square.

  1. Results

Uncertainty measures, effect sizes and accurate representation of results is lacking!

  1. Discussion
  • Please use the aforementioned references to back up the discussion.
  • Please add more on the potential value of this study.

Minor editing comments:

  1. Intext references need reformatting. The numbers should be placed in square brackets but not as superscripts.
  2. In the first paragraph of Introduction and throughout the manuscript, please replace “people” with “population.”
  3. Too many run-on sentences and confusing sentences are used, which need extensive editing throughout.

Author Response

Dear Reviewer,

Thak you for your insightful comments on our manuscript. We feel the comments have helped us significantly improve the manuscript.

Point 1: Title: Title of the study does not reflect its design. Authors may consider adding information about the study design.

Response 1: Indeed, as you pointed out, the wording was meant to confuse the reader. We changed the title to “Attitudes and behaviors toward COVID-19 vaccination in Japanese university students” and clearly stated that the purpose of the study was to grasp the actual situation of university students by comparing their attitudes and behaviors after vaccination with those of faculty members. ( Lines 1, 19-24, 86-91)

Point 2: Introduction: The introduction section not back up by sufficient literature review and is superficial. In other words, it does not cover the entire breadth of the issue at hand. Relevant references can be added to strengthen the evidence related to the factors associated with vaccine hesitancy. Tesfaye et al., in one of the articles (given below) emphasize on vaccine literacy and vaccine confidence index. Authors are encouraged to include some information related to the uptake of other protective behaviors e.g. handwashing and mask wearing by the younger population

Response 2: Thank you for your valuable comments and information. We have tried to enhance the Introduction by adding literature on vaccine hesitancy, as well as an article on infection prevention behavior. (Lines 63-71, 95-102)

Point 3: Methods: and Response 3:

First sentence needs to be re-written. : We have corrected the points you pointed out. (Lines 112)

Please re-write line # 67: We re-wririted. (Lines 116-120)

Please add the date of ethical approval: We added that. (Lines 132)

Please move the ethical information into a separate paragraph and label it as “Ethical Considerations.” In addition to the Ethical approval, what others measures authors have taken to ensure the ethical conduct of the study?: We moved the ethical information into “Ethical Considerations”. (Lines 129-136). And we added about ethical considerations. (Lines 133-136)

How authors prevented multiple responses from the same participant?: We added an explanation of how to prevent multiple responses. (Lines 125-127)

Which platform was used to collect the electronic data?: The survey was conducted within a dedicated line for members of the university.

Was the survey tool validated? It is critical to use valid instruments?: We added that the questionnaire was constructed by collecting the opinions of internists, psychiatrists, public health experts, university administrators, and faculty members. (Lines 151-153)

Statistical analysis portion does not provide the information on the power analysis and type of statistical tools other than Chi-square.: We added description of the power Analysis. (Lines 168-171)  No statistical tools other than the chi-square test are used.

Point 4: Results: Uncertainty measures, effect sizes and accurate representation of results is lacking.

Response 4: As you indicated, we have added the missing points. (Lines 195, 211, 212, 214, 215, 217)

Point 5: Discussion: Please use the aforementioned references to back up the discussion. Please add more on the potential value of this study.

Response 5: We have added the content of the Discussion along with the references. (Lines 240-243, 314-330)

Point 6: Minor editing comments and Response 6

  1. Intext references need reformatting. The numbers should be placed in square brackets but not as superscripts.: We corrected the style of references.
  2. In the first paragraph of Introduction and throughout the manuscript, please replace “people” with “population.”: We replaced “people” with “population”.
  3. Too many run-on sentences and confusing sentences are used, which need extensive editing throughout.: Thank you for pointing this out. By correcting the points you have pointed out so far, We have improved the text to be more logical.

Round 2

Reviewer 3 Report

Reviewer comments for authors

Thank you for the opportunity to re-review the manuscript titled “Attitudes and behavior toward COVID-19 vaccination in Japanese university students”

  1. Title: It is improved now. Just add “A cross-sectional study” after the students separated by a colon.

  1. Introduction

Thank you for adding the suggested references. However, I think mention the geographical area of the study is important. For example, Yadate et al., in the United States.

  1. Methods

Thank you for addressing methodological comments.

  1. Results

Authors are encouraged to add tables and figures. At the minimum, please add table 1 of descriptive statistics.

  1. References 9 and 27 are duplicated. Please replace ref # 27 with the actual one which is given below: https://pubmed.ncbi.nlm.nih.gov/34682953/

Thank you and all the best!

Author Response

Dear Reviewer,

Thank you for your valuable suggestions. It has helped us to refine our paper.

Point 1: Title: It is improved now. Just add “A cross-sectional study” after the students separated by a colon.

Response 1: Thank you for your valuable comments. We have added the title as you indicated.

Point 2: Introduction: Thank you for adding the suggested references. However, I think mention the geographical area of the study is important. For example, Yadate et al., in the United States.

Response 2: We have added the geographical information as you indicated (Lines 79-81).

Point 3: Results: Authors are encouraged to add tables and figures. At the minimum, please add table 1 of descriptive statistics.

Response 3: We added Table 1 that indicated a number of subject population.

Point 5: References: References 9 and 27 are duplicated. Please replace ref # 27 with the actual one which is given below: https://pubmed.ncbi.nlm.nih.gov/34682953/

Response 5: Thank you for pointing out the error in the literature. We have corrected it as you indicated.